# Providing Housing First services for an underserved population during the early wave of the COVID-19 pandemic: A qualitative study

Cilia Mejia-Lancheros[1,2]*, James Lachaud[1], Evie Gogosis[1], Naomi Thulien[1,3], Vicky Stergiopoulos[4,5], George Da Silva[1], Rosane Nisenbaum[1,3,6], Patricia O'Campo[1,3], Stephen Hwang[1,3,7]

1 MAP Centre for Urban Health Solutions, Li Ka Shing Knowledge Institute, St. Michael's Hospital, Unity Health Toronto, Toronto, Ontario, Canada, 2 Research Group in Nursing Care and Practice, Family Health Nursing and Health Measures; Nursing Faculty, Universidad Nacional de Colombia, Bogotá, Colombia, 3 Dalla Lana School of Public Health, University of Toronto, Toronto, Ontario, Canada, 4 Centre for Addiction and Mental Health, Toronto, Ontario, Canada, 5 Department of Psychiatry, University of Toronto, Toronto, Ontario, Canada, 6 Applied Health Research Centre, Li Ka Shing Knowledge Institute, St Michael's Hospital, Unity Health Toronto, Toronto, Ontario, Canada, 7 Division of General Internal Medicine, Department of Medicine, University of Toronto, Toronto, Ontario, Canada

* Cilia.Mejia-Lancheros@unityhealth.to

## Abstract

### Objective

We assessed the critical role of Housing First (HF) programs and frontline workers in responding to challenges faced during the first wave of the COVID-19 pandemic.

### Method

Semi-structured interviews were conducted with nine HF frontline workers from three HF programs between May 2020 and July 2020, in Toronto, Canada. Information was collected on challenges and adjustments needed to provide services to HF clients (people experiencing homelessness and mental disorders). We applied the Analytical Framework method and thematic analysis to our data.

### Results

Inability to provide in-person support and socializing activities, barriers to appropriate mental health assessments, and limited virtual communication due to clients' lack of access to digital devices were among the most salient challenges that HF frontline workers reported during the COVID-19 pandemic. Implementing virtual support services, provision of urgent in-office or in-field support, distributing food aid, connecting clients with online healthcare services, increasing harm reduction education and referral, and meeting urgent housing needs were some of the strategies implemented by HF frontline workers to support the complex needs of their clients during the pandemic. HF frontline workers experienced workload

**Data Availability Statement:** The study data cannot be made publicly available due to the sensitive nature of the data (small study sample

with sensitive details, which could lead to participants and their clients' identification), the agreements and procedures governing using the collected dataset established by the study sponsor, research institution and Research Ethics Board, and the data usage and sharing, and the privacy protection assured to the study's participants in the consenting process. However, de-identified participant data used in the present study may be made available to investigators who complete the following steps: 1) present a study proposal that has received approval from an independent research committee or research ethics board; 2) provide a data request for review by the study principal investigators (PIs) and co-investigators; 3) following approval of the request, execute a data-sharing agreement between the investigators and the Study PIs. Study proposals and data access requests should be sent to Katherine Francombe Pridham (Katherine. Francombe@unityhealth.to), the research manager for this research study.

**Funding:** The present study received funding from St. Michael's Hospital Research Training Centre Scholarship awarded (2019) to Dr. Cilia Mejia-Lancheros. The funding institutions had no role in the study design, collection, analysis and interpretation of the data or the preparation, revision, or approval of the present manuscript. The views expressed in this publication are the views of the authors.

**Competing interests:** The authors have declared that no competing interests exist.

burden, job insecurity and mental health problems (e.g. distress, worry, anxiety) as a consequence of their services during the first wave of the COVID-19 pandemic.

## Conclusion

Despite the several work-, programming- and structural-related challenges experienced by HF frontline workers when responding to the needs of their clients during the first wave of the COVID-19 pandemic, they played a critical role in meeting the communication, food, housing and health needs of their clients during the pandemic, even when it negatively affected their well-being. A more coordinated, integrated, innovative, sustainable, effective and well-funded support response is required to meet the intersecting and complex social, housing, health and financial needs of underserved and socio-economically excluded groups during and beyond health emergencies.

## Introduction

Millions of people have been infected and died due to the SARS-CoV-2 virus (COVID-19) worldwide [1]. Prevention, containment, and mitigation actions/plans (e.g., testing and contact tracing, social distancing and isolation/quarantine, sanitation and lockdown) have been put in place in different countries to curb the spread of COVID-19 and limit adverse health-derived outcomes for populations (e.g., mortality rates) and health systems (e.g., the collapse of acute healthcare services) [2–5]. Intersectoral collaboration between health, social and community-based organizations is considered a critical strategy to appropriately respond to health emergencies such as the ongoing COVID-19 pandemic [6, 7]. Academic, health, community-based organizations, and the private sector have joined efforts to provide healthcare services, personal protective equipment (PPE) and medical supplies, and support services to respond to the COVID-19 pandemic and reduce its associated adverse health, economic, and social effects at the individual and societal levels [5, 8].

Local community-based participation is pivotal in responding to and implementing effective and timely responses to tackle the adverse health and socio-economic consequences of the COVID-19 pandemic [9, 10]. Such engagement of local organizations is particularly relevant to providing and enhancing appropriate social and health supports to underserved populations such as those with lived experiences of homelessness or unstable housing [9–11]. Structural factors such as poverty, social exclusion, housing affordability, racism, colonialism, discrimination, and stigma intersect with homelessness, poor health and well-being [12–15]. Therefore, socio-economically excluded populations, such as those experiencing homelessness with or without concurrent and intersecting issues (i.e., employment and skills development barriers, mental and physical illness, discrimination, food insecurity, powerlessness, social disconnections, lack of access to appropriate health supports and social opportunities) suffer the most adverse social and health outcomes during public health emergencies [11, 16–18]. They are more likely to live in precarious, unsafe and overcrowded environments, face barriers to accessing appropriate healthcare and social benefits, and are less likely to seek help if infected [16–20]. Therefore, these populations have a greater risk of adverse health, economic, and social impacts as a result of the ongoing COVID-19 pandemic and other public health or social emergencies. These structural barriers may also contribute to non-adherence to social public

health measures (e.g., social distancing and quarantine) and medical recommendations/treatments during the pandemic events and beyond [16–20].

Organizations that provide support services (e.g., access to food, housing, and social support services) to people with current and past experiences of homelessness and those experiencing poverty and complex mental health and social needs, play an important connecting and trusting role for their clients [21, 22]. Therefore, they are critical actors in planning and implementing contingency and mitigation responses to prevent and reduce the spread of infection outbreaks and the adverse social consequences among socio-economically excluded people [9, 10]. Despite such a vital pandemic response role, the organizations supporting people with past or current experiences of homelessness/housing instability and mental illness already faced significant pre-pandemic challenges and barriers such as lack of spatial, human, economic, and material resources [23]. This may further diminish their capacity to appropriately support and address the complex needs of people who use these services during public health emergencies. Furthermore, frontline workers of these organizations are at high risk of exposure to the virus if no appropriate safety and protection measures are adopted and followed, as well as enough financial and personal resources being mobilized [24]. Additionally, miscommunication, lack of concrete actions and leadership, and poor coordination [25, 26] among local community-based actors and with local and national authorities could weaken the ability to respond to and address the complex social, health, and economic needs of socio-economically excluded people during and post public health emergencies and beyond [27, 28].

Few studies have documented the response of programs serving people with a history of chronic homelessness/housing instability, mental illness and those experiencing poverty and social exclusion during the COVID-19 pandemic. Likewise, no studies have assessed the adjustments implemented by Housing First (HF) programs that provide immediate access to housing and mental health supports without preconditions [29, 30] during the first wave of the COVID-19 pandemic and the challenges, experiences, and lessons learned. Understanding the response role, strengths, weaknesses and needs of such organizations during public health emergencies is critical to building and integrating further the local pandemic response preparedness to ensure that both clients and frontline providers of social, health, housing and other support programs get the appropriate, timely and equitable support during such health events so that the pandemic's adverse effects be effectively mitigated. Hence, the present qualitative study seeks such evidence through the following objectives:

1. To identify the challenges faced by HF programs and frontline workers to deliver support services to people with experiences of homelessness and mental illness during the first wave of the COVID-19 pandemic.

2. To characterize the strategies implemented by HF programs and frontline workers to overcome the main experienced pandemic-related challenges and meet their client's social, housing, and health needs.

3. To explore the impact of the COVID-19 pandemic and associated public health and social response measures on the well-being (health, social and work) of HF frontline workers during the early wave of the pandemic.

4. To map the main recommendations that HF frontline workers suggest to strengthen or improve the support services delivered to underserved and socio-economically excluded people during public health emergencies and beyond.

## Material and methods

### Study design, setting and population

The present qualitative study is a sub-study embedded within the "Toronto At Home/Chez Soi: Qualitative follow-up study" (TORONTO AH/CS-QUALI STUDY) (https://maphealth. ca/ahcs-quali-study/) carried out from April 2020 to October 2020. The TORONTO AH/ CS-QUALI STUDY aims to explore service user and provider perspectives on the circumstances, challenges, experiences, and unmet needs that may have prevented HF clients from achieving better social, health, and other recovery-oriented outcomes while receiving long-term HF services through the Toronto site of the At Home/Chez Soi study carried out during 2009–2017 [31]. This COVID-19-focused sub-study explores provider perspectives during the first wave of the pandemic, as the implementation of the Toronto AH/CS-quali study coincided with the beginning of the COVID-19 pandemic in Canada. We amended the TORONTO AH/CS-QUALI STUDY protocol to include the COVID-19 sub-study and received approval from the Research Ethics Board (REB) at Unity Health Toronto, Toronto, Canada. The qualitative methodology can offer a valuable perspective to explore the impacts of the challenges, opportunities and lessons learned in providing support and response services to underserved or socio-economically excluded populations during the COVID-19 pandemic [32].

For the purpose of the sub-study, we recruited 20 HF clients and nine HF frontline workers in Toronto. The recruitment and main COVID-19 findings from HF clients have been published elsewhere [33]. For the present study focused on the perspectives and experiences of HF frontline workers, we targeted and recruited a purposive sample of nine HF frontline workers from the three first and largest HF programs (run by three community-based organizations: Across Boundaries (AB), COTA Health and Toronto North Support Services (TNSS)) that implemented HF services for people with lived experiences of chronic homelessness and severe mental disorders in Toronto, Canada, back in 2009 under the implementation of At Home/ Chez Soi study in Toronto [31, 34]. The target sample (N = 9) included three HF frontline workers Case Managers(CMs) for each of the three HF agencies was decided prior to the study based upon the following factors.

First, although the three agencies deliver HF services, there are differences between them based on the type of clients served and the core HF services approach they deliver. AB provides HF with Intensive Case Management (ICM) with specific services oriented toward ethno-racial populations with moderate needs level (MN) for mental health services; TNSS provides HF with ICM for clients with MN for mental health services regardless of their ethno-racial background; COTA provides HF services with Assertive Case Management (ACT) for clients with high need level (HN) for mental health services.

Second, HF Case managers (our study populations) provide frontline HF services in a client/Case Manager ratio of 20:1 for HF with ICM and 10:1 for HF with ACT. More detailed information on HF-specific services (ICM and ACT) delivery by the targeted sample and the three agencies can be found in a previous publication by Hwang et al [34].

Third, we were interested in having frontline workers with a longer-term history (between 3 and 10 years) of working as CMs within the three HF programs to better answer the objectives of the TORONTO AH/CS-QUALI STUDY, which go beyond the COVID-19 pandemic. Considering all this, three CMs from each HF program (N = 9) was sufficient to answer our study objectives and to observe potential differential and common perspectives, experiences, challenges and strategies to help us have more comprehensive and generalizable findings on the studied topics.

Our study sample was recruited using a written invitation letter containing the rationale, goals, objectives, methodology, benefits and potential harms, principal investigators' contact details, and the ethical approvals of the study. Letters were circulated by email through HF program managers, as well as by asking the first interviewed participants to be free to invite their colleagues to participate in our study (snowball sampling). Interested individuals contacted the study co-principal investigator and lead author (CPI: CML) or the study's peer research assistant (PRA, GD). The PRA or the CPI subsequently contacted all HF frontline workers who expressed their willingness to participate to confirm their participation in the study and in the virtual semi-structured audio-recorded interview.

## Study's ethics

The TORONTO AH/CS-QUALI STUDY, including the COVID-19 sub-study, received ethics approval from the REB at Unity Health Toronto, St. Michael's Hospital, Toronto (Canada). All participants received a written consent form containing information regarding the study rationale and objectives, methodology, type of data to be collected, and potential benefits and harms of the study. Participants could provide oral or written consent to participate in the study, have their interviews audio-recorded and transcribed, and use de-identified interview quotations in public or academic knowledge translation outputs. The oral consent process was documented in writing by the PRA on the consent form just prior to the virtual interview. All participants received an electronic gift card with a value of CAD 40 for their participation in the study despite being frontline workers in the midst of the pandemic.

## Data collection

The qualitative semi-structured interviews were conducted virtually using Zoom for Healthcare [35], the virtual platform approved by the Unity Health Toronto REB to conduct research interviews. We used a flexible interview guide comprised of semi-structured-COVID-19 related questions to address the study's COVID-19 objectives. After each interview, the two facilitators had a debriefing session to discuss salient topics, identify discourse saturation, or examine new areas that needed further in-depth exploration in subsequent interviews. The interview guide is presented in S1 Table in S1 File. The individual interviews lasted between 35 to 45 minutes. To further enhance the privacy and security of the data collected, we recorded interviews using an external audio-recording device rather than using the internal audio-record feature of the Zoom platform. The collection of rich and comprehensive narrative data at each interview was facilitated by two experienced study team members: PRA (a self-identified man with lived expertise in social exclusion, injustice, mental illness and housing instability) and either the CPI (a mixed-raced scientist self-identified as a woman) or study's co-investigator (a scientist self-identified as a man from black ethno-racial background). Both scientists had expertise in homelessness and health, socio-economic determinants of health, and health inequities/inequalities.

## Data analysis

We conducted the data analysis process manually (an Excel file was used to organize and facilitate the coding process), but using an interactive, dynamic and participatory approach to enhance the trustworthiness, validity, and reliability of the data analysis process and derived findings [36]. Our analysis was guided by the Analytical Framework (AF) method for multidisciplinary-qualitative studies [37] and thematic analyses [38].

First, if needed, we de-identified (removed any name that may have led to participants' or their close colleagues or clients' identification) audio-recorded data collected from participant

interviews. Second, the de-identified audio-recorded data were transcribed into text format by an experienced external transcribing service and verified by at least one study team member. Third, all transcribed text data were carefully read line-by-line by the study's lead author and CPI (who also participated as a co-facilitator in most interviews) for a first in-depth familiarization with the data.

Fourth, from the nine interview transcripts, a sample of three transcripts was identified as having richer information by the lead author and PRA. The three selected transcripts were read in-depth by three members of the research team (including the team members who facilitated the interviews) to identify common codes and develop a coding and themes framework. The framework follows a tree-style and is composed of third-, second and first-grade themes as identified from the coding process. The third-level themes (or third-level categories) were the lower or basic themes yielded from the identification and analyses of a cluster of identical or similar coding keywords (e.g. challenges in supporting the social and health needs of clients in-house, in-person services resulting from suspension of the residential or in-person mental or social support services codes). The second-level themes were derived from the identification and analyses of cluster or alike third-level themes (e.g. challenges in delivery in-person support services, derived from the challenges in supporting the social and health needs of clients in-house, in-person services themes). The main or first-level themes were derived from the further identification and analyses of alike main relevant second-level themes that linked to the study objectives (e.g., Challenges that HF frontline providers faced in serving HF program clients' theme were informed by the second-level themes: challenges in delivery in-person support services, challenges in delivery in -agency-related services, or provide appropriate virtual support services).

Fifth, after each of the three members independently coded and categorized the data of the three-interview transcript sample, the CPI reviewed the resulting preliminary coding to confirm common codes or add new codes and themes, therefore completing the core code/theme AF. Sixth, this initial AF was further validated in an interactive working session by four members of the research team, including the lead author, the two members who participated in the initial coding and categorization of the three-interview transcripts, and one of the researchers who did not participate in the initial coding. Seventh, the validated AF was then applied to the remaining six interview transcripts by three research team members (each member applied the AF to two interviews). During this process, each member could identify and add new codes or second and first themes that were not presented in the existing AF.

Once the coding and theme identification process was finalized, the CPI read the remaining six interview transcripts and validated the inclusion of newly identified themes and categories in the AF (very few additional second-and-third themes emerged), completing, therefore, the final AF, which included salient quotations and preliminary interpretations made by the coding team. Finally, we analyzed and interpreted the coded and themed data, guided by a mix of inductive and deductive approaches [39] using the thematic analyses [38] and social-ecological model (particularly the updated motives and influence: proximal processes sub-model) [40, 41] and critical theory [42] as references. The CPI conducted the first analysis and interpretation of the AF, followed by a validation process by three other members of the team and critical revision and further validation by the rest of the paper's co-authors.

## Results

Study participants were nine HF frontline adult workers (six self-identified females and three males: two females and one male for each agency) who worked as Case Managers in the three HF organizations and provided HF services to people experiencing homelessness and mental

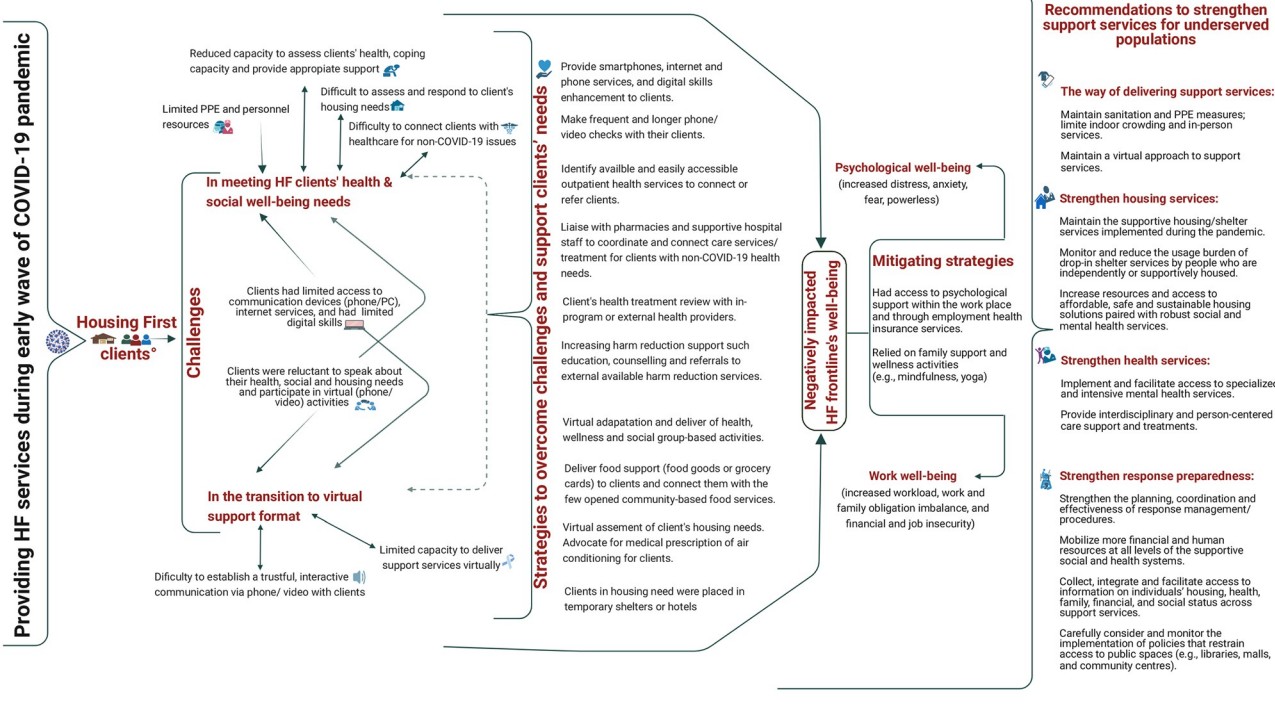

**Fig 1. Main identified themes and their direct and indirect relationships.**

disorders in Toronto. Most participants (90%) had worked with the same HF programs for more than three years before the interview date and continued to work (virtually or in-person (only in urgent cases)) with HF clients over the COVID-19 pandemic until the interview date.

The main themes were identified (with direct or indirect relationships) and linked to study objectives as presented in Fig 1 and expanded below. In summary, all identified main themes (the HF services response challenges, strategies implemented, effects on frontline workers' well-being and associated coping strategies, and the recommendations) are interconnected despite each theme or sub-theme having its particularities. This shows the complex, interrelated and multidimensional impacts of the COVID-19 pandemic on HF support response capacity and adaption during the first wave of the pandemic in Toronto, Canada. They also show the burden of COVID-19 as a disruption event and the response role of HF frontline workers on their well-being.

## The challenges faced (Objective 1)

With the sudden onset of the COVID-19 pandemic and implementation of strict public health measures (lockdowns, mandatory quarantines, physical and social distancing), the HF programs and in particular the HF frontline workers, found themselves facing several interrelated challenges in providing the habitual and urgent HF supports to their clients, as they were not prepared to provide a quickly appropriate and effective response. The challenges that were experienced were grouped into the following two main categories.

## Challenges in meeting HF client's health and social well-being needs

During the pre-pandemic, HF workers regularly checked HF clients' health, social status, and needs in person, either in the HF agencies facilities, at the client's home, or in community settings (e.g., coffee shops and parks). Furthermore, HF clients also had the option to actively participate in group-based social activities (e.g., social dining, healthy social talks, physical and psychological wellness activities, and social networking events). However, at the beginning pandemic, all of these activities abruptly stopped and quick changes needed to be implemented, such as transitioning to virtual support formats (video or phone-based) usually from the home of the HF frontline workers; therefore, suspension of all non-urgent in-person encounters between HF personnel and clients were implemented.

*"It is a challenge. We're so used to the face-to-face contact and you know talking to them and then you know when you visit them, they might say, can we go for a coffee and we take them for a coffee you know; there's a lot of things that you can't do. They can't even come into the office as a drop-in. We can't do that anymore."*

*[B0006]*

These changes and intersecting critical issues, such as limitations in the availability and access to technology and communication resources (e.g., not all clients had access to mobile or computer devices and phone or internet services, see more on the below theme), personal protective equipment (PPE), personnel resources (e.g., not enough personnel to attend to clients' urgent needs in person) and clients' health issues triggered several barriers for HF workers to respond to their client's needs in an appropriate and timely way.

HF frontline workers expressed challenges in assessing the coping capacity of their clients and changes or exacerbation in their mental or physical health issues, particularly among those with severe mental illness, substance use, mobility limitations, risk of food insecurity, as well as clients with very limited or non-social ties outside of those they had developed as a result of their involvement in HF programs. This made HF frontline workers feel powerless regarding the amount, quality and frequency of the support they could provide to meet such clients' needs.

*"We're doing on phone calls and also sometimes WhatsApp, usually once a week so we can see if there is a mental health concern. If you know the person is okay with the video and because not everybody is okay with the video, it is a challenge"*

*[B0001C].*

Adding to this, especially during the first months of the pandemic, HF workers also had difficulty assisting clients with non-COVID-19 health issues to connect with healthcare services, as the health system was overwhelmed and many health services were suspended or shifted to virtual visits.

*"The reality is, is it is, everything is very limited. Everything is taking a lot longer-you know, doctors' appointments. As you know, [it is] hard to get things done. It has become quite a bit different. Everyone is trying to do things virtually, and that is a problem for our clients/"*

*[B0003C].*

Similarly, in the first months of the pandemic, HF frontline workers had a hard time assessing the housing needs or housing status of some of their clients, mainly those with limited access to a phone or other communication device/service and those living in non-shared or single residential units. They were unable to contact them by phone or in-person.

*"I would say the biggest challenges are kind of our social activities and community visit. We do face-to-face visits in the community. Those are very much limited at this time. I can't do a home visit; right, that's like the most core part of my job 'cause that's why I'm hired; right."*

[B0008]

This increased their worries about the lack of clients' ability to maintain healthy and safe home environments, address housing issues (e.g., water leaks or appliance repairs, air-conditioning malfunctioning) and adhere to residential unit rules (e.g. reduced noise) which could impact their housing tenure. Furthermore, HF programs needed to suspend the housing relocation services for clients, as in-person visits to rental units were suspended and there were not many rental housing options available.

## Challenges in the transition to a virtual support format

The shift to implementing a virtually-based HF support approach to ensure HF clients continued to receive some amount and type of HF services/support during the first wave of the pandemic brought different challenges for HF frontline workers. Among such challenges was the limitation in delivering appropriate phone or video-call-based services/support to their clients, as many of their clients did not have smartphones and PCs or access to internet and phone services in their residential units.

*"I had to call my clients. Some of them didn't have phones, some of them didn't have Wi-Fi. I had to you know do a virtual visit. So, no technology, no phones, no Wi-Fi. I think technology and Wi-Fi, Internet, these two have been very, very important, you know like problems"*

[B0009]

Another challenge experienced by HF frontline workers was that some clients were unwilling to establish a trustful, interactive and informative communication process via phone or video, which eroded the existing active and interactive communication relationship they had built with their clients before the pandemic.

*"It's not the kind of relationship building that we have usually in the past; and our clients, they're experiencing it, they're not really voicing it but you can tell they're reacting differently to the way we are right now."*

[B0003C]

Furthermore, even when some clients started to access and use virtual support services/activities, the HF frontline workers noticed that their clients reacted differently or interacted little on the phone or during virtual support sessions compared to during in-person active or collaborative interactions they had before the pandemic.

*"I think it's a bit harder to build rapport and that trust when it's just, if you're, you know, calling them over the phone or just seeing them over a screen."*

*[B0008]*

Thus, HF frontline workers were concerned that these challenges were reducing the continuity of the support process, undermining the social, health and housing progress and improvements that some clients had accomplished before the pandemic.

## Strategies to overcome challenges and support clients' needs (Objective 2)

Within the financial, material, personnel and functioning constraints experienced over the first months of the COVID-19 pandemic, HF programs and HF frontline workers implemented different strategies to provide as much support as possible to their clients with greater social, food, housing, and health needs and to provide some kind of continuity of pre-pandemic support services and activities either with punctual in-person visits (adhering to public health measures) or virtually over the phone or PCs.

### Strategies to support health and social well-being needs

HF frontline workers knew some of their clients could experience an onset or exacerbation of their mental and physical health problems due to mandatory lockdowns, quarantines, and isolation/social distancing, as well as the stopping or significant reduction in existing social and health support services within and outside HF programs; therefore, they came up with several strategies, which seemed to work well. For example, HF frontline workers implemented more frequent and longer phone checks (sometimes daily or two or three times a week) with their clients, especially those they knew had severe or complex mental or physical disorders(including substance additions) and were or not in treatment before the pandemic, as well as with those clients, they were able to identify evident changes or decline in their mental or psychological status.

*"We do make a lot of phone calls to our clients and try and speak to them and ask them how things are going. Sometimes, it's difficult to talk about certain things on the phone. I think having them see us would be a lot better, but we just can't at this point in time; right."*

*[B0003C]*

Also, they identified available and easily accessible outpatient healthcare services (e.g., family doctors or nurses, psychiatrists), and liaised and advocated with local pharmacies and supportive hospital staff (i.e., hospital-based social workers) to coordinate and connect care services for their clients with need for non-COVID-19 care or treatment. Nevertheless, such care was delivered in its majority virtually. In-person care was only available for severe health issues or clients with assisted treatment (e.g., IV pharmacotherapy).

*"I know some of the GPs have done like over-the-phone appointments at this time, so we will support them as well with that. The loaner phones had been helpful for that because we can provide a phone for them to connect with their GP and have an assessment done that way. I have collaborated with the management in trying to phone the person[client] and then the doctor."*

*[B0008]*

Furthermore, HF frontline workers updated their clients' residential, personal and medical history. They also closely revised treatment adherence and doses of those clients with the evident onset or exacerbation of mental disorders. Frontline workers followed up with HF program health providers (psychiatrists, nurses, physicians) or primary care physicians to assess and implement treatment adjustments if required. This strategy seems to improve clients' health and assist HF frontline staff in planning continual supports for those with greater health needs.

*"We are trying to advocate and say actually he is not doing well doctor and set up and advocate for medications. It's very important. We talk to them [physician] so they [clients] can get the medication."*

*[B0001C]*

As HF frontline workers observed an exacerbation or worsening of mental health problems (including substance use including alcohol) among some HF clients, they were able to increase some support services via phone or virtually, such as harm reduction education, connecting to in-program counselling specialists, and referrals to external (in-person or virtual) harm reduction services to which their clients could turn to if needed.

*"We're doing a lot of addiction kind of you know support and connections. And we are doing more harm reduction education and support [resources] for that. We are able to access crisis, Gerstein crisis centre, and we are also connected and work with CAMH [Centre for Addiction and Mental Health]. We collaborate with them very well. So, that is the access that we provide and refer them to*

*[B0001C]*

After the first three months of the pandemic, some of the HF programs and the HF frontline workers also started virtual health-related stress/uncertainty relief and socializing sessions (e.g. peer-led conversations about welfare, medication, and health issues, weekly virtual yoga and mindfulness sessions) to further support the social and health issues experienced by their clients.

*"Peer support is actually a separate program online now. Conversations, issues about medical, medication, health issues, so you have peer support and discussion in groups, we're doing that now."*

*[B0001C]*

HF frontline workers reported that although such virtual activities started with few clients, they observed that more clients were connecting or interested in such activities.

Furthermore, since many social support services that provide food and meals to their HF clients were closed as a result of the COVID-19 pandemic, putting some of their clients at high risk of food insecurity, as many clients continued to live in poverty or had reduced financial capacity to meet their food needs. Thus, HF programs organized small food baskets and grocery gift cards for clients with greater food needs. Items were delivered to clients either by HF frontline workers (for clients with physical mobility issues or limited transportation services) or by clients picking it up at HF program facilities.

*"We're always making sure that they have food and if they don't, we get it for them, or we ask for it. My manager goes out and buys them some stuff sometimes as well."*

[B0006]

HF frontline workers also updated their clients regularly on the very few free meal programs and food services available in their neighbourhoods, which mainly provided takeaway options. However, many services that continued to operate were far away from the client's home, thereby limiting access for many clients.

*"There were a few places where a person could get a meal, and we would direct them to those places, but they were very far from where most of our clients live."*

[B003C]

### Strategies to enhance the transition to the virtual support approach

HF programs and frontline workers set up several strategies to enhance the adaptation to the sudden shift from in-person to virtual support due to the pandemic. Among such strategies, there was a partnership with a local telephone service provider to give free smartphone devices and provisional internet and phone connection services to HF clients who did not have access to electronic devices or internet services.

*"Some of them [clients] didn't have a phone, some of them didn't have Wi-Fi. I had to you know, do a virtual visit. So, no technology, no phones, no Wi-Fi. So, the program, you know, [we] tried to resolve this, and we gave our clients phones which is great."*

[B0009]

Nevertheless, such a digital and communication initiative was challenging to implement as some clients lacked knowledge of the functioning of virtual systems or smart devices, and others did not want to use them. Thus, HF frontline workers provided in-person or phone help/ guidance on the basic functioning of the given phones, set up email accounts and explained how to navigate the virtual services HF programs and other support platforms provided.

*"Now the challenge is with the phones. It's too short-sighted to just give somebody a phone that's never used a smartphone. They just do not [the phone functioning), so what I try to do is sit down with them for an hour and go over like hey, this is your emergency contacts, and like this is how you pick up a phone call, keep it simple."*

[B0007].

Despite the above, most workers reported that giving phones and access to the internet and phone services was very beneficial, as it helped them to reach and support more clients virtually. In addition, many clients could also engage in virtual services offered within and outside HF programs and connect with friends, relatives and acquaintance-based social networks.

### Strategies to support housing needs

Though the majority of their HF clients were stably housed (have independent or assisted place to live), HF frontline workers put in place various housing-related strategies to prevent

the loss of accommodation, enhance housing stability, and resolve housing issues arising during the pandemic lockdowns or stay-at-home orders. For example, for clients with videophone services, HF frontline workers regularly performed wellness checks to ensure the state, cleaning and maintenance of their clients' accommodation. HF frontline workers also regularly asked their clients about any housing issues (either physically or administrative such as renewal of renting contracts) that needed to be fixed and either arranged assistance through the HF program or landlords. These strategies were reported to work well.

*"I say, I just want to check if you're doing okay, and I just see, you know, If the landlord is taking care of fixing anything broken. I just want to see the unit, how they're doing: can I have you on video? Then I tend to just look at that [residential unit]."*

[B0001C]

As the ongoing pandemic and strict public health response measures in Toronto continued over the summer of 2020, the lack of air conditioning was an important issue for many HF clients who did not have access to air conditioning services in their home due to limited financial resources. In addition, they could not access most public spaces with air conditioning that they used to do over the summer previous to the pandemic (e.g., libraries, coffee shops, malls) as they were closed or had very restricted access. Thus, HF frontline workers requested financial support from the HF program or a medical prescription for air conditioning, particularly for clients with underlying health problems, as a physician could prescribe air conditioning as part of a patient's treatment plan. Hence, clients could apply for financial support to deliver and install a cooling device such as an air conditioner or fan as part of their treatment coverage. However, HF frontline workers highlighted that it was not a smooth process and required time, as it implies a lot of bureaucratic steps. Also, not all physicians knew or were willing to prescribe air conditioning as part of a treatment plan.

*"If you have a doctor that can write really well [the need for air conditioning] for you, then you can get the air conditioning. If he doesn't, if it's not good, then you don't get it. That's how it has been and it's still a challenge."*

[B0001C]

Finally, secure and affordable housing became much harder to obtain during the COVID-19 pandemic. If private market housing could not be obtained for new clients or those who had scheduled a housing change or end of their housing lease, as well as clients who required COVID-19 mandatory quarantine but were living in congregate settings (e.g., boarding home or rooming house), HF programs would house such clients in temporary shelters or hotels funded by the local government during the pandemic until a new home was found or the clients completed the mandatory quarantine period.

*"We are now getting temporary housing which could, in the form of some hotel rooms, basic hotel rooms, but they're available for our clients. If there's a situation where our client has to leave their place, or they get taken into our team and they have to find a place we're putting them up in some hotels for a little while. We are accepting new clients, but most of those people will probably be placed in a hotel until things cool down for a bit and those hotel rooms are supplied by mostly the government or, you know, funders of our agency as well."*

[B0003C]

## Effects on the HF frontline providers' well-being (Objective 3)

The following main identified themes give a unique glance at the negative effects that COVID-19 and associated responses had on the psychological, work, and socio-economic well-being of HF frontline workers who were actively providing their services during the first pandemic wave. However, our participants felt they received great support/resources within and outside their work organizations, which positively aided in their coping and overcoming such effects.

### Psychological well-being effects

A great amount of fear and worry of being exposed to COVID-19 was a commonly manifested psychological effect among HF frontline workers, especially those engaged in providing in-person support to their clients. Although they used PPE, they felt vulnerable to contracting the COVID-19 virus because some clients did not entirely adhere to social distancing and stay-at-home orders (mainly due to mental impairment) or could not afford PPE (e.g., masks and hand sanitation due to lack of financial resources or supply shortages).

> *"As a case manager, I think we are facing with a lot of problems, fear, because I don't know, where these clients, you know, have gone [around] and which people they meet. Some of my clients really like you know to see or their friends and go everywhere."*
>
> *[B0009]*

High psychological stress and anxiety were identified as other frequent psychological issues experienced by HF frontline workers. However, participants recognized that such effects were mostly related to their feelings of powerless and inability to appropriately assist those clients with more health and social support needs (e.g., clients whose mental health worsened or who were at high risk of overdose, self-harm, and suicidality) and the new scenario where they found working alone and finding solutions to meet their client's needs.

> *"I would say not being able to see your co-workers, mentally it's affecting me [more stress and anxiety], but also [not] being able to consult them, just having people on-hand at the office [where] I can say, hey, I need your help with this new scenario or with this thing popped up."*
>
> *[C0007*

Many of the HF workers frequently questioned their supportive role and felt unprepared, undertrained, and under-resourced to provide an effective and timely response to clients with complex intersecting needs and those with less ability to adapt to the new virtual way of pro-viding support services.

> *"I was trying to help my client as much as I could so, and like some of them, they didn't have phones, no technology, no like Wi-Fi. Then, you know, I heard this news [client's death from overdose], so it was very difficult for me to put you know, everything in place, put myself together and you know, deal with this like tragic news."*
>
> *[C0009]*

## Work well-being effects

The increased workload was a frequent issue, negatively impacting HF frontline workers. Adapting to working remotely from home and pivoting to a virtual environment increased the overall hours that HF frontline workers spent providing support to clients. It also challenged their ability to plan meaningful support activities virtually but also organize and deliver safe in-field support services for clients in most need. Further, working from home was affecting their capacity to have a positive balance between work and family obligations, as work was taking priority over personal or family activities/obligations (e.g., parenting and schooling children).

> *"Working from home, unfortunately, is much harder in the sense that there is no downtime. I'm working seven hours straight at home; it's a lot. You're doing a lot more in that time. 9:00 to 5:00 is like the expectation, but now during COVID, we're allowed to work from 8:00 to 8:00, so there's a little bit of flexibility we have, but sometimes I do leave my phone on a little bit later and I keep [working] on."*
>
> *[C0007]*

Other work-related effects included the feeling of increased work and financial insecurity that some HF frontline workers were experiencing, as they feared that they might lose their job due to agency closures or a reduction in the many support services implemented due to the pandemic.

> *"I could see many stressors, right? Like my son, he's four years old, and his daycare was closed, I had him at home. And I didn't know, you know, what happened with my job, so [if] we continue with this agency or we have to go on like government support programs and things like that. So, I feel somehow it [insecure and worry]."*
>
> *[C0009]*

## Strategies to mitigate the effects on the HF frontline workers' well-being (Objective 3)

HF frontline workers adopted several strategies or had access to support services, which helped them cope with the above psychological and work-related struggles. All participants shared that they received psychological support from their work environments. For example, they could access psychiatric services or psychological counselling within their work environments and receive psychological support from co-workers and managers through debriefing sessions, which was a space to share the everyday work effects on their psychological well-being.

> *"It's definitely been a stressful time, but I feel like I have been, you know fairly well supported by my team and yeah. I think that is helped us keep going through this pandemic; I think you know, we're all kind of adjusting and trying to find the new normal."*
>
> *[B0008]*

Some HF frontline workers also had access to psychological support through external resources such as employee health benefits, which provided phone-based professional counselling over the pandemic, thus helping them unpack feelings of distress, fear and anxiety.

*"In addition to that, the agency has insurance that is part of the work insurance policy that we could [use] to talk to the employee's counselling services. You can call in or you can go in [person], but now, it's calling and get the support that you need."*

[B0001C]

All HF frontline workers had family or relatives as their primary support source of support outside their work setting. Additionally, most HF frontline workers started to adopt self-care approaches to boost their wellness, such as balancing their work and family responsibilities, practicing meditation or mindfulness, and prioritizing themselves and their family's well-being. However, some frontline workers recognized that it required time to see their effects.

*"I really tried hard by mindfulness technique to try you know, to calm myself down and cope with this situation, you know; but it took some times."*

[B0009]

## Recommendations to strengthen the support services (Objective 4.)

Reflecting on both pre-pandemic and ongoing pandemic issues, challenges, and experiences encountered as HF providers working with people experiencing homelessness, social exclusion and mental illness, and the response strategies implemented within HF programmes and beyond, our participants elucidated some recommendations to further strengthen the supportive response during and beyond public health emergencies.

### Recommendations for the way of delivering support services

HF frontline workers believed that some public health and social measures implemented to contain the COVID-19 pandemic should be maintained during the post-pandemic period. Among the first group of measures recommended are those related to health protection, such as having fewer clients within shelters or supportive programs' facilities, having access to PPE equipment for staff and clients, and reducing the frequency of in-person encounters between clients and support workers in indoor or confined spaces. In addition, some HF frontline workers considered that these measures could contribute to reducing the spread of other infectious diseases among support workers, clients and local communities.

*"You know it's going to be better practice the social and public health measures for us to always maintain a good healthy distance from people."*

[B0003C]

Maintaining a virtual approach to support services was highly recommended by almost all participants. This allows workers to reach more clients in need of housing, social, and health support and help the workforce be more efficient in allocating services and tailored activities according to their client's needs, including facilitating access to a speedy referral within and across social, health, and housing systems. Nevertheless, participants recognized that this approach could lead to more isolation and loneliness among some clients, which may add to other significant support services disparities. Some clients face financial barriers that leave them without access to PCs/smartphones or internet or phone services.

*"Technology; offering the free phones, stuff like that. You know more mental health supports virtually, and more virtual programs are all free. I would hope that these continue because they have pretty immense success but do I think it will continue after? No, I think a lot of it will be halted. Maybe there might be a hybrid or lessening of it, but, I believe some of the virtual programs will always remain but a lot of other services might be completely gone later."*

[B0007]

## Recommendations for strengthening housing services

HF frontline workers believed that COVID-19-related measures implemented within the supportive housing and shelter system should be maintained and strengthened beyond the ongoing pandemic. Among such measures are improved safety and sanitized conditions, reduced dormitory capacity and increased spacing within shelter clients.

Moreover, some HF frontline workers suggested that effective and long-term strategies to reduce the usage burden of drop-in services by people who moved to safe independent or supportive housing units without the need to do so should be implemented and monitored. This would allow other people without access to stable housing and supports to get a bed/room in the shelter system and optimize limited housing resources.

*"It will definitely have permanent changes to how we deliver with clients because we know that our clients sometimes use the shelter system even though they have housing; right! And we have some issues, some people who are like that and who are you know, want to be near their friends, so their friends may be in a shelter, and they'll go and see [stay with] them."*

[B0003C]

Finally, all HF frontline workers called for local, provincial and federal governments and associated stakeholders to increase resources and access to affordable, safe and sustainable housing solutions paired with robust social and mental health(including access to food) support for people experiencing homelessness/housing instability, as doing so would reduce other socioeconomic and public health concerns such as poverty and crime.

*"You know, I believe everyone needs access to the very basic needs. It doesn't matter, I don't care about barriers, you just got to get people a place to sleep that's safe and to eat. If you don't have those two things, you're going to have crime, you're going to have poverty, you're going to have these overlapping barriers that people can't get through. It's just the basic human, human right and need, I believe that you need access to food, housing."*

[B0007]

## Recommendations for health-related services

HF frontline workers highlighted that the pandemic exposed the lack of adequate nurse and physician resources in the community to cope with the complex physical and mental health needs of underserved or socially excluded populations during the pandemic times and beyond. Therefore, they asked for increasing or mobilizing robust financial investment and resources to expand and retain healthcare personnel supporting these populations.

*"Nurses and doctors worked hard, really super hard and it shouldn't be that way. We should have those services you know. So, taking care of your healthcare is a preventive measure, so you need to have more hospitals, more facilities, more trained staff, more doctors, more nurses."*

*[B0005]*

Also, HF frontline workers recommended increasing access to more personalized, appropriate, timely, integrated and sustainable healthcare services during and beyond public health emergencies. Among such suggestions is the implementation of and facilitating access to specialized and intensive mental health services to appropriately meet the mental care needs of people experiencing homelessness, as many of the existing support services do not cover such needs. As many emphasized that such population groups frequently experience complex and intersecting mental health and social needs that require interdisciplinary and person-based supports and treatments to better enhance their health recovery and overall well-being.

*"I think also just more availability for mental health services like different therapies because, I mean you know, we can provide this, you know a limited amount of counselling and the medication piece; but also, I think a lot of people out there have [mental disorders], you know, they need more intensive therapy."*

*[B0008]*

## Recommendations for response preparedness

Despite social support programs showing a remarkable capacity for problem-solving and prioritizing some essential services during the first wave of the COVID-19 pandemic, HF frontline workers found multiple gaps that should be addressed in order to build or strengthen the preparedness to better provide timely and coordinated (social, economic, health, food, and communication) support services for clients of HF and alike support programs and for people experiencing homelessness without having supportive services.

Among such strategies, there is a need to strengthen the planning, coordination and effectiveness of response management and procedures; mobilize more financial and human resources at all levels of the supportive social and health systems (e.g., housing/shelter, mental and additional support services, food support services, technological and social connection services, skills development or training, and social networking connections, and internet and phone services).

*"Also, just economically[support], like maybe more employment programs generated towards the people that have mental health or addictions issues, more activities, more you know, social opportunities-so many things, yeah."*

*[B0008].*

Another strategy that was suggested was to provide more efforts and mechanisms to collect, integrate and facilitate access to information on individuals' housing, health, family, financial benefits, and social networking status across supportive services need to be implemented. This would increase opportunities to deliver timely, appropriate and coordinated care for people in most need during public health emergencies and beyond. Furthermore, it would reduce the

bureaucracy and the need for clients to share personal information with service providers, thus ensuring the continuity and personalization of care and social support.

*"Any type of support you need immediately can be delayed significantly, and we learned that. For instance, it's much harder to reach ODSP. People are most vulnerable; unfortunately, they can't manage some aspects of their lives. I am [cheeked] who are my contacts, I have their right phone number or the right pharmacy phone number. So, I'm realizing the importance of having these set up. I think our organization did a good job in the sense of like well we need to update our [clients] profiles, in full. Especially during this pandemic because we need to know how to best serve them in any type of emergency situation."*

[B0007]

Finally, some HF frontline workers suggested that any potential policies that seek to restrain access to public spaces (e.g., use of libraries, malls, and community centres) should be carefully considered and assessed, even during public health emergencies. Many of these public spaces constitute the only areas where underserved and socio-economically excluded people can break their isolation and loneliness, satisfy heating and cooling needs, and access much-needed resources.

*"Yeah, the world is going to be very small for our clients. Libraries, parks, community centres, malls. All these things are gone. And that's another thing too like say, you were going to the malls and people will be rushing them out now more because of stigma or you know, all kinds [things]. I definitely think they're going to change, definitely, they're going to be feeling more isolated and things are going to be less available to them."*

[B003C]

## Discussion

The present qualitative study carried out in the first wave of the COVID-19 pandemic in Toronto (Canada) is one of the few to reveal the challenges faced and response strategies implemented by three large HF programs and their HF frontline workers to meet their clients' needs. It also revealed the burden caused by the pandemic as a stressor event and the support response role of HF frontline workers in serving HF clients.

In our study, we found that despite the important role that HF programs have as a means of ending homelessness [31, 43], they faced different challenges to deliver appropriate and timely support to their clients in the first months of the onset of the COVID-19 pandemic and implementation of strict public and social responses. Among such challenges was the suspension of in-person or in-field supports within the HF programs and across external health, housing, food and social services and the shift to a virtual (phone or pc-based) approach. This led to a reduction in the capacity to assess and respond to the needs of HF clients, particularly in providing continuity of social and health support for clients with a high need for mental health, food, social and housing support. Also, it reduced their inability to detect the decline or onset of mental health disorders (including substance use) and provide appropriate care either through HF programs or external healthcare providers.

Other research has examined the main challenges faced by homelessness services/programs (including HF) in high-income contexts during the COVID-19 pandemic and found similar findings to our study. For instance, a pilot UK study carried out over 12 weeks

during the first wave of COVID-19 found similar findings to our study [44], especially regarding disruptions that HF staff experienced in providing face-to-face support and communicating with their clients using virtual channels [44]. Findings from another study of community-based organizations supporting people experiencing homelessness in the USA, found similar challenges to those we observed, such as the inability to provide continued support for non-COVID-19 needs and lessening of the pre-pandemic built relationships between outreach and clients [45]. This was mainly due to the reduction of in-person support, the take-over of the virtual approach level, and the limited availability of personnel and material/ financial resources for appropriate and timely response to clients' multidimensional needs during the pandemic [45].

Our findings complement findings from existing literature describing the financial, material and human resource constraints that organizations face in responding to the complex and intersecting social, health, housing, economic and food needs of underserved or socially excluded population groups during and beyond the COVID-19 pandemic. As Parkes et al highlighted, support services/programs, "already operated like it was a crisis, because it always has been a crisis" [46]. The growing homelessness crisis [47] associated with often under-resourced, limited, inequitable and poor integrated/coordinated support services [48–50] jeopardizes their support mission in both usual and pandemic times, putting socially excluded or underserved populations in deeper pathways of exclusion, illness, poverty, social disconnection, chronic homelessness and diminished recovery opportunities.

Notwithstanding the big challenges, we found that HF programs and their frontline workers implemented several strategies to overcome the challenges and contribute to addressing their clients' health, housing, social and food needs. One of the main strategies was to quickly shift some of their in-person support services (follow-ups on mental and physical healthcare and housing needs, delivering a social connection to group-based discussion or well-being activities, such as yoga) to a virtual format. This strategy was boosted by the provision of a smartphone and internet and phone connection to clients who did not have access to such services through a partnership with a local telecommunication provider. Also, for clients with very specific and complex health needs, in-person or in-field services were provided when necessary (e.g., administration of non-oral medicine, renewal or signing of tenant contracts, delivery of food goods); however such services were not continuous but rather implemented as needed. Other studies reported similar response strategies implemented by homeless-serving organizations in other high-income settings, such as in the United Kingdom [46, 51] and the USA [52]. Our study and others provide evidence of the critical role that community-based organizations, their frontline workers [45], and successful partnerships with the private sector [53, 54] and other agencies [55] have in innovating and quickly adapting support services to respond to the needs of underserved or socially excluded populations during public health emergencies.

However, it is important to emphasize that although the strategy of moving many support services to a virtual format was a useful means of ensuring some kind of continuity in the basic support for HF clients and other underserved or socio-economically excluded populations during the early wave of the pandemic, there were some clients that did not benefit from such adjustment. Some clients could not navigate the virtual support system due to mental health impairment or lack of digital skills in using smart communication devices and video connections, or because the virtual approach failed to build or maintain trustful relationships between clients and support workers. Thus, these digital inequalities [56, 57] do not only constitute a further barrier for the socio-economically excluded to access and receive appropriate social,

housing and healthcare support, it may also worsen their feelings of social isolation/disconnection and self-harm [50, 58].

Another innovative and unique strategy that we identified in our study was that HF frontline workers advocated for the medical prescription of air conditioning for those clients with severe medical conditions and a lack of air conditioning in their homes. However, it was not easy and timely to accomplish. Despite the prescription of an air conditioner being covered ($1,000) by the Canadian government [59], it was still difficult to get such a prescription from healthcare providers. This may be because healthcare providers are not familiar with the rights and importance of prescribing non-pharmacological interventions as part of the individuals' health treatment plan.

Finally, as a response to the observed increase in substance use, exacerbation or onset of mental disorders and self-harm behaviour among some HF clients, HF workers tried to increase the provision of harm reduction support virtually (e.g., harm reduction education, increase counselling with program mental health workers) or facilitate referrals to external support. Yet, it was not enough to cope with the needs due in part to the pandemic response (closure of or limited opening times of in-person substance use and harm reduction services) and the lack of harm reduction preparedness and reediness at the programming and system levels to cope with the surge of mental and social problems (e.g. isolation/lockdowns) or to ensure substance treatment continuity during first months of COVID-19 pandemic, in Toronto and across other Canadian jurisdictions [60, 61]. Therefore, effective, coordinated, scalable harm reduction response planning should be strengthened to address substance use needs and prevent and mitigate associated harm across public emergencies and beyond. For example, maintaining open (scale them if needed) supervised substance use facilities as essential health services during crises has shown to have greater life and health benefits than closing them during the COVID-19 pandemic in Australia [62]. In Dublin (Ireland), improving access to methadone, benzodiazepine and naloxone treatment, including home delivery of prescription drugs (e.g., methadone and benzodiazepine) and implementation of new inpatient units for rapid initiation to opiate substitution therapy for people with suspected or positive COVID-19, appeared to contribute to low COVID-19 infection and mortality rates among people experiencing homelessness and substance use issues [63].

Regarding the impact of COVID-19 on HF frontline workers' well-being, we found that HF frontline workers faced significant burdens of psychological distress and an array of feelings such as powerlessness, overwhelming, fear, insecurity, and anxiety. Moreover, they experienced unpreparedness and felt untrained to provide appropriate levels of support for their clients during the COVID-19 pandemic and feared job loss due to the reduction or closure of many social support services. These findings are in line with a Canadian survey carried out among frontline workers who provide social support to youth with experiences of homelessness during the first wave of the COVID-19 pandemic [64]. It found that the wellness of support staff was negatively impacted by COVID-19, as they experienced high levels of burnout, work-related stress, compassion fatigue, and high workload due to the increased needs of people that required support services during the pandemic [64].

Similar to our study, other studies also found that the pandemic and the associated adverse socio-economic and health effects caused on the general public, but particularly to underserved, mentally ill, lonely and socio-economically excluded populations, inflicted a high amount of psychological and physical stress, fatigue and onset of mental disorders, the decline in health status, job insecurity, and discrimination on frontline workers supporting such population groups [44, 45, 65]. COVID-19 pandemic should be considered a serious traumatic stressor event [66], which could have both short and long-term mental health consequences

for both clients of social support services and frontline providers [65]. To help cope with these impacts, all our HF frontline workers received great and continuous psychological and occupational support within and from their employers and HF program mental health professionals, thus contributing to mitigating psychological and practical work-related struggles. External sources of support, such as employee assistance programs, health benefit plans, family members, and well-being activities (e.g. yoga and meditation) were also available to and accessed by our HF frontline workers. Few studies have reported that a lower percentage (< 24%) of social and health support workers sought in-person (< 8%) or online (23%) mental health support from professional or formal sources during the 2020 COVID-19 pandemic [65], contrary to a high percentage that relied on less formal sources of support such as family and friends [65, 67].

These findings suggest that it is critical to implement and enhance access to informal and formal sources of social and mental health and wellness support services for frontline workers serving vulnerable people during pandemic events and beyond [45, 64, 67]. This can contribute to building resilience, enhancing overall and occupational-related health well-being and recovery, and mitigating the risk of work loss and economic insecurity among support services workers.

Our HF frontline workers suggested key recommendations to strengthen HF and non-HF support services during the ongoing COVID-19 pandemic and the post-pandemic recovery period in four areas: ways to deliver social support, supportive housing services, healthcare services, and health emergency response preparedness. Maintaining and enhancing the physical distancing policies and safety when in-person support services are delivered to their clients and the flexibility to continue providing support services remotely were seen as important strategies to retain over the long term. In addition, maintaining the COVID-19 implemented sanitation, physical distance, and safety measures across the shelter and support system could reduce risk exposures and the spread of infectious diseases within marginalized and socioeconomic populations well beyond the current public health emergency states [64, 68]. Yet, long-term public health measures must be carefully balanced with other initiatives, such as increasing safe and affordable shelter and supportive housing, social material, financial and personnel resources [69, 70]. Otherwise, they may constitute a further barrier to people's access to supportive services, including access to safe shelter/housing [55].

On the other hand, virtualization of social and health support services could have their "losers" and "winners". As for some underserved or socio-economically people, it can open more and better opportunities for timely access to a variety of support services from health to work or training services, contributing to more holistic and personalized support ("winners") and having more likelihood of life recovery [45]. For others, it may contribute further to amplifying their vulnerability, exclusion and inequalities, making it harder to exit homelessness, and poverty and recover their health and social well-being ("losers") [45]. Illness, preferences or lack of communication device/services, could lead people with intersecting and complex social, housing, and health needs to experience further social and health vulnerability and inequities. Therefore, sustainable, equitable and long-term digital and in-person support approaches are both needed to ensure and facilitate the underserved and socio-economically excluded people [57, 64] to reduce the risk of letting some of them out of the recovery train due to digital inequalities [56, 71].

Besides recommending immediate access to safe and permanent housing, which is critical to reducing outbreaks and pandemics among unstably housed populations [50, 72], HF frontline workers suggested the need to find appropriate strategies to reduce the double usage burden of the temporal or transitional nature of shelter facilities by those who had already transitioned to housing. Despite this double usage behaviour seeming to be frequent, there is

scant evidence of its magnitude and driving factors. People who transition from homelessness to housing require comprehensive support services to help them in the transitioning process, such as strong health, social connection, recreational, friendship, and everyday functioning support that meet people's needs and help them adapt to new housing and life pathways [73].

Finally, alongside increasing coordination, integration, and collaboration across social and health support services, our study participants recommended that more specialized mental health services (including substance use and harm reduction) and health and social support professionals (well-paid and cared supportive workforce) along with more personalized socio-economic supports are required to strengthen and improve the support responses for underserved populations during and beyond pandemics. Lessons learned from the COVID-19 pandemic should support allocating and investing additional (financial, material and personnel) resources across social, health and housing systems to ensure a more inclusive, equitable, sustainable and innovative post-pandemic recovery plan for socio-economically excluded and underserved populations [52, 55, 72, 74–78].

## Limitations

Despite being one of the few studies assessing the impact of COVID-19 on the delivery of HF services to people with lived experience of homelessness in a large, urban and high-income country, the present study has the following limitations. The data collection was done during the first wave of the COVID-19 pandemic (April to July 2020); thus, it is likely that HF programs have implemented further strategies and responses to support their clients with less challenging and more client engagement. It may also be that HF frontline workers have experienced new challenges in the service delivery process as the COVID-19 pandemic and associated public health and social measures continue to be in place in the study setting. Furthermore, the response capability of HF programs to address their client's needs during the pandemic could have differed from that of other supportive organizations serving underserved population groups, which may have had either more or limited financial, material, and personnel resources. Similarly, this study's findings might differ from those in other geographical contexts with different socioeconomic conditions and political and emergency response capacities. Therefore, it should be considered when extrapolating or comparing the present study's findings.

## Recommendations for programming, policy practice, and research

In Table 1, we provide key recommendations for strengthening the role of social, health and housing organizations or programs serving underserved populations during and beyond local or global public health emergencies.

In conclusion, despite the several work-, programming- and structural-related challenges experienced by HF frontline workers when responding to the needs of their clients during the first wave of the COVID-19 pandemic, they played a critical role in meeting the communication, food, housing and health needs of their clients during the pandemic, even when it negatively affected their well-being. A more coordinated, integrated, innovative, sustainable, effective and well-funded support response is required to meet the intersecting and complex social, housing, health and financial needs of underserved and socio-economically excluded groups during and beyond public health emergencies. Ensuring appropriate work, health and financial support for support services workers will further contribute to appropriately meeting the needs of underserved populations.

**Table 1. Recommendations for HF programs, government, community and for research, and practice.**

| Targeted area | Key recommendations |
|---|---|
| **HF programs and alike services** | • Evaluate the short and long-term support response during the COVID-19 pandemic to identify areas of improvement, innovation and adaptation opportunities for strengthening their emergency response management, readiness and capacity during both pandemic and non-pandemic times.<br>• Assess the engagement and effectiveness of virtual support services implemented during a pandemic to determine their long-term sustainability.<br>• Consider hybrid and peer-led group-based support service formats to enhance inclusion and engagement based on clients' accessibility to technological resources or in-person preferences.<br>• Consider support services diversion (e.g., cooking classes, digital skills enhancement, virtual health navigation, and virtual/remote work opportunities) according to clients' needs and lessons learned during a pandemic to enhance further their financial, social, leisure, and health recovery.<br>• Assess the opportunities to implement permanent and safe harm reduction physical facilities and in-person harm reduction services (e.g., methadone, naloxone, benzodiazepine provision) within their support programs for clients in need. Otherwise, advocate for and coordinate a facilitated and easy access to such existing services within clients' residential communities.<br>• Advocate for permanent access to affordable internet and phone services for program clients, and underserved and excluded populations to enhance their access to the appropriate social, training, employment, leisure, housing, financial and health services.<br>• Strengthen inter-agency, private, community and government partnerships to increase support response integration and coordination to address clients' needs in an effective and timely manner during crisis and non-crises times.<br>• Create training opportunities for support staff at all levels to further strengthen their skills and capabilities and leadership to effectively and safely support the intersecting complex needs of support services clients during emergencies and everyday support services.<br>• Provide a variety of wellness support services and benefits, including mental healthcare, psychological counselling, adequate work-related resources, salary and work stability for frontline and other support staff. |
| **Local community** | • Advocate for affordable and safe housing options for unstably housed individuals within local communities.<br>• Actively engage in creating inclusive and equitable community-based activities and spaces to enhance social integration and participation for residents from non-white ethno-racial or cultural backgrounds, those living alone, or experiencing housing instability, poverty, unemployment and mental and physical illness.<br>• Strengthen the support for local food banks and meal programs to prevent and address food insecurity among underserved or low-incomes population groups through local partnerships between community-based organizations and local business. |
| **Governmental level** | • Invest in permanent, affordable and safe housing options for underserved or socio-economically excluded populations.<br>• Expand HF-based services paired with a strong portfolio of employment, training, financial, recreational and health services to support people who have experienced either temporary or chronic homelessness.<br>• Increase specialized mental health services, including substance use and self-harm reduction supports and make them available, easy and safely accessible to individuals and communities in need. Considerer innovative and sustainable partnerships with existing-community-based programs providing social supports to expand mental health, substance use and self-harm reduction services through their well-stablish programs (e.g., HF, supportive housing).<br>• Based on existing evidence-based and lessons learned over the pandemic, create effective, innovative, integrated and coordinated support response systems and services to enhance a more equitable and sustainable crises response and recovery for underserved and socio-economically excluded individuals, families, and communities.<br>• Increase and sustain the financial support and resources allocated to existing social, housing, and health organizations working with underserved or socio-economically excluded populations to strengthen their usual and crisis support capacity. |
| **Research and practice** | • Assess the long-term challenges and effectiveness of social, housing and health support (including substance use and harm reduction) responses implemented during the COVID-19 pandemic.<br>• Evaluate the long-term impact of the COVID-19 pandemic on the mental, physical and social health needs of frontline workers, considering the different socio-economic and political contexts.<br>• For health professionals, consider non-pharmacological treatments, such as prescription of air conditioning as part of the healthcare plan of individuals with health conditions.<br>• Offer safe virtual or in-person healthcare services based on the client's health and resources needs rather than on health providers' desires. |

# Supporting information

**S1 File. Individual interview guide.**
(DOCX)

**S1 Checklist. COREQ (COnsolidated criteria for REporting qualitative research) checklist.**
(PDF)

# Acknowledgments

We thank HF frontline workers for their participation and wiliness to share their experiences during the first wave COVID-19 pandemic. We also thank the coordinators and directives of the three HF agencies, which facilitated contact with study participants and supported and encouraged their participation during their paid working hours.

# Author Contributions

**Conceptualization:** Cilia Mejia-Lancheros.

**Data curation:** Cilia Mejia-Lancheros, Evie Gogosis.

**Formal analysis:** Cilia Mejia-Lancheros, James Lachaud, Evie Gogosis, George Da Silva.

**Funding acquisition:** Cilia Mejia-Lancheros, Stephen Hwang.

**Investigation:** Cilia Mejia-Lancheros, James Lachaud, George Da Silva.

**Methodology:** Cilia Mejia-Lancheros, James Lachaud, Evie Gogosis, George Da Silva.

**Project administration:** Cilia Mejia-Lancheros, Evie Gogosis.

**Resources:** Cilia Mejia-Lancheros, Stephen Hwang.

**Software:** Cilia Mejia-Lancheros.

**Supervision:** Naomi Thulien, Vicky Stergiopoulos, Rosane Nisenbaum, Patricia O'Campo, Stephen Hwang.

**Validation:** Cilia Mejia-Lancheros, James Lachaud, Evie Gogosis, Naomi Thulien, Vicky Stergiopoulos, George Da Silva, Rosane Nisenbaum, Patricia O'Campo, Stephen Hwang.

**Visualization:** Cilia Mejia-Lancheros.

**Writing – original draft:** Cilia Mejia-Lancheros.

**Writing – review & editing:** Cilia Mejia-Lancheros, James Lachaud, Evie Gogosis, Naomi Thulien, Vicky Stergiopoulos, George Da Silva, Rosane Nisenbaum, Patricia O'Campo, Stephen Hwang.

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
