## [Decision Letter · Decision Letter 0]

22 Mar 2022

PONE-D-21-25281Providing Housing First services for an undeserved population during the COVID-19 pandemic: A qualitative study.PLOS ONE

Dear Dr. Mejia-Lancheros,

Thank you for submitting your manuscript to PLOS ONE. After careful consideration, we feel that it has merit but does not fully meet PLOS ONE’s publication criteria as it currently stands. Therefore, we invite you to submit a revised version of the manuscript that addresses the points raised during the review process.

We look forward to receiving your revised manuscript.

Kind regards,

Saeed Shahabi

Academic Editor

PLOS ONE

2. When reporting the results of qualitative research, we suggest consulting the COREQ guidelines: http://intqhc.oxfordjournals.org/content/19/6/349. In this case, please consider including more information on the number of interviewers, their training and characteristics.

“The present study received funding from St. Michael’s Hospital Research Training Centre Scholarship awarded (2019) to Dr. Cilia Mejia-Lancheros. The funding institutions had no role in the study design, collection, analysis and interpretation of the data or the preparation, revision, or approval of the present manuscript. The views expressed in this publication are the views of the authors.”

“The present study received funding from St. Michael’s Hospital Research Training Centre Scholarship awarded (2019) to Dr. Cilia Mejia-Lancheros. The funding institutions had no role in the study design, collection, analysis and interpretation of the data or the preparation, revision, or approval of the present manuscript. The views expressed in this publication are the views of the authors.”

“The authors declare no competing interests”

6. We note that you have indicated that data from this study are available upon request. PLOS only allows data to be available upon request if there are legal or ethical restrictions on sharing data publicly. For more information on unacceptable data access restrictions, please see http://journals.plos.org/plosone/s/data-availability#loc-unacceptable-data-access-restrictions.

Reviewers' comments:

Reviewer's Responses to Questions

**Comments to the Author**

1. Is the manuscript technically sound, and do the data support the conclusions?

Reviewer #1: Yes

Reviewer #2: Yes

2. Has the statistical analysis been performed appropriately and rigorously? 

Reviewer #1: Yes

Reviewer #2: Yes

3. Have the authors made all data underlying the findings in their manuscript fully available?

Reviewer #1: Yes

Reviewer #2: Yes

4. Is the manuscript presented in an intelligible fashion and written in standard English?

Reviewer #1: Yes

Reviewer #2: Yes

5. Review Comments to the Author

Reviewer #1: This qualitative study examined the experiences of service providers working in Housing First (HF) settings in Toronto, Ontario, Canada. This analysis is a sub-study from a rigorous, ongoing parent study of the long-term impacts of HF using both quantitative and qualitative methods. The manuscript offers insights into what it is like to work in HF programs during the pandemic, including the challenges, strategies implemented to support service users, impacts on providers, and lessons learned. Its novelty lies in its focus on the HF service provider experience for which, at this point in the pandemic, there are no known studies of HF providers – a perspective that was also somewhat under-examined prior to the pandemic. Further, this is an essential workforce that has been working throughout the pandemic (despite fears of becoming unemployed, according to this study), often in the shadows on the more traditional healthcare system. Accordingly, this study offers some validation of this sector’s hard work, including the challenges of the job. The manuscript has potential to be an important source of information on what this workforce experienced early on in the pandemic in Canada and offer valuable insights for the recovery to come, as well as future public health emergencies. However, there are several issues that would first need to be addressed. First, the major issue here is the length of the results section, which is particularly cumbersome. Its current structure also leaves more to be desired in terms of a synthesis that would allow readers to efficiently extract key findings (described further below). As such, the results need to be condensed substantially (less is more here) and the overlap in a number of themes presents a feasible path to doing this. Second, much of the discussion involves a summarization of the findings. It would be more useful to focus on how the findings fit with other related research (suggestions made below), highlight the implications of key findings, speak to future avenues for research, and devote more space to discussing the policy recommendations. Third, the study design requires more of a rationale as to why it was selected (not the study’s qualitative methods, but its recruitment and sample size). These comments are explained in more detail below, along with a number of minor points.

Abstract

- The conclusion needs more of a ‘so what?’ element

Introduction

- Not familiar with the term “Equity-seeking groups,” can guess its meaning, but please define (further, particularly important to make clear who you’re referring to given the next sentence with notes serious consequences)

- The case for the value of the research can be strengthened in the final paragraph of the introduction; objective 4 gets at this a little bit, but it needs to be clearer what understanding “the challenges, experiences, and lessons learned” during wave 1 of the pandemic will do for the HF sector/research community. The first two sentences on why qual. methods are valuable can also probably be scrapped, or moved to methods at the very least.

Methods

- It’s a small sample drawn from 3 organizations. This is not necessarily a weakness of the study, though a rationale for why data collection was set or stopped at 9 participants is certainly needed. Providing a justification of 3 organizations is probably warranted as well, as opposed to recruiting 9 participants from a single organization, for example.

- Very minor: make clear that the sub-study received REB approval and that it is not just the parent TORONTO AH/CS-QUALI STUDY

- Kudos for providing a good value honorarium to HF workers (it’s not common enough practice when conducting research on providers and this was a workforce, which was dealing with a barrage of pandemic-related issues and feeling undervalued when the study was conducted, so this was a considerate decision). No action needed on this comment.

- “To further enhance the privacy and security of the data collected, we recorded interviews using an external audio-recording device rather than using the internal audio- record features of the Zoom platform.” Poor transcriber(s). No action needed on this comment.

- Can probably remove the word “solid,” which is a little on the casual side (though I have no doubt the scientist did have a solid research background in the noted areas).

- Describe what was the “initial tree-styled AF”

- Define first-, second-, and third-grade themes, and their hierarchy (if applicable)

Results

- Is there any more data available on the characteristics of the sample? For example, gender, ethnicity, age, education, et cetera. Also, provide range for participants’ HF work histories: “All HF frontline workers had worked with HF clients for more than three years.”

- Opening quote of theme 1: “The reality is, is it is, everything is very limited…” Might the participant have said “as it is”?

- The results and their thematic structure are LONG (by my count: the results are currently 24.5 pages, excluding figure 1). Their length presents two key problems: [1] There is overlap and redundancies throughout the results section and [2] It’s hard to tell what is most important and how the findings between first-grade themes intersect. As an example of the former, sections 1.1 and 1.2 both discuss the challenges caused by phone communication (e.g., “…many clients were reluctant to disclose their mental health needs by phone…” in section 1.1 and “HF frontline workers shared that many of their clients were reticent or did not want to speak about personal matters or health issues via phone or virtually” in section 1.2). As an example of determining what is most important, the third-grade themes in the second main theme describe 21 different strategies used by providers. However, commentary on what worked well, what did not, what it was like for providers to implement the strategies, what was perceived as most important are not often commented upon. Further, what might be innovative (e.g., requesting a medical prescription for air conditioning – very interesting, would like to know more about how providers found this, extent to which they felt this was effective) gets lost in the mundane (e.g., this foundational component of case management: “They were also constantly advocating with key liaison actors (e.g., social workers, nurses) in primary care and hospital-based care services to ensure the access and delivery of appropriate care, including hospital admissions and continuity of care for non-COVID-19 related issues.”). These issues make me wonder if the analytic approach had a sufficient synthesis built into it, as there remains a lot of data presented descriptively.

- Minor: Revise for clarity, as alcohol consumption is a type of substance use (“in response to the increased substance use, and alcohol consumption”)

- Typo: 3.3 coping strategies

Discussion

- The discussion is mostly focused on summarizing the findings, with a sprinkling of contextualization using recent research. However, there are some omissions that could be considered by the authors (prioritizing other qualitative studies in the articles suggested below):

Kaur et al. (2021). Provision of services to persons experiencing homelessness during the COVID-19 pandemic: A qualitative study on the perspectives of homelessness service providers. Health Social Care in the Community

Levesque et al. (2021). Understanding the Needs of Workers in the Homelessness Support Sector. Homeless Hub report.

Peters et al. (2021). A systematic review and meta-synthesis of qualitative studies that investigate the emotional experiences of staff working in homeless settings. Health Social Care in the Community (not specific to pandemic, though some of your findings match up with this)

Pixley et al. (2021). The role of homelessness community based organizations during COVID‐19. Health Social Care in the Community (not specific to HF, though some of your findings match up with this)

Parkes et al. (2021). ‘They already operated like it was a crisis, because it always has been a crisis’: A qualitative exploration of the responses of one homeless service in Scotland to the COVID-19 pandemic (not specific to HF, though some of your findings match up with this)

Another grey evidence source is CAEH’s COVID-19 information exchange (https://caeh.ca/homelessness-sector-covid-19-information-exchange/), which provides evolving data on the impacts of the pandemic on the homelessness/housing service sector in Canada. It would be interesting to see if what they were hearing from providers across the country during the first wave (again, more broader scope than only HF providers) is consistent with your sample. Food for thought.

- The first limitation is NOT a limitation and needs to be removed. It is invalidating toward the experiences of HF workers and the authors have not suggested that the findings represent HF service users, so the point is irrelevant.

Practice and Policy Recommendations

- Clarify that “social housing programs” do not refer to “social housing”

- The recommendations are fairly comprehensive and general. Three considerations on which I would encourage the authors to reflect: [1] Do the recommendations overstep the findings? [2] Is a qualitative study of 9 participants sufficiently positioned to make such sweeping recommendations? and [3] What meaningful actions can HF programs take to support clients and staff during the pandemic? (note: HF programs are not resourced and/or positioned to make some of the transformative changes that are recommended here; what are the small, but potentially significant changes that HF programs could implement)

- There are a number of typos in Figure 2

Other

- Be consistent with language for HF recipients through the paper (i.e., clients or service users)

Reviewer #2: The manuscript “Providing Housing First services for an underserved population during the COVID-19 pandemic: A qualitative study”, examines the experiences of housing first providers during the first wave of the pandemic in Toronto, ON, Canada. This manuscript is well-written, the findings are interesting and have important implications for the provision and implementation of Housing First. There are a few minor adjustments I would recommend. The Results section should be refined more to highlight the most important themes. The implications of these central findings could be developed more in the Discussion. The Discussion should more clearly convey the authors' core conclusions.

Results

In the Results section the reader is directed to Tables 1-4. These tables are somewhat difficult to read and interpret. The authors should consider either removing the tables and re-integrating excerpts in the text, and/or revising the tables so that they are less packed with text and easier to interpret. It should be easier for the reader to make the link between the excerpt and theme, and to understand the excerpts.

A little more signposting is needed throughout the Results. For example, on p. 11 I was confused to discover that “Challenges that HF frontline providers faced in serving HF program clients during the COVID-19 pandemic” was just the overarching theme, and that the authors would be instead presenting the sub-themes in detail. Make clear this is an introductory paragraph and not a substantive theme (and throughout the remaining sections too).

This leads me to another issue, if including an excerpt, it needs to be followed by an explanation, so either explain here and throughout the Results, or include with interpretation in another place in the text. For example, on p. 12 I’d like to hear more about the excerpt, why was virtual a problem for clients? On p. 27, a supporting excerpt in the section on “High psychological stress and anxiety” would go a long way.

I really like Figure 1. and wonder why it isn’t integrated better in the results or Discussion sections. The authors should consider this as a basis for developing their overarching framework.

Themes are plentiful, maybe some are more important or hold stronger implications for practice than others? I would suggest that the authors consider reducing or integrating themes to highlight the most important ones. For example, very important themes include the strategies implemented to support HF clients’ housing needs, and the implementation of housing-related adjustments.

Commentary on “implementation of housing-related adjustments” on p. 33 should be expanded particularly around reducing the reliance on drop-in shelter services for people in housing. This is a common issue in the field but not represented in the literature to-date, as far as I know.

Themes read as categories that map onto the research objectives rather than as themes representative of patterns in the data. Adjust theme names so they tell the reader how they inform the research objectives.

Minor comments, suggestions

There are some interesting findings this reviewer would like to hear more about, on p. 22 how did frontline workers increase harm reduction education and support? And were their advocacy efforts successful? P. 25 more discussion on the provision of temporary accommodation in the form of homeless shelters/hotel programmes. P. 29, really interesting that frontline workers accessed support through employment support programmes, this is not available to HF workers in many other contexts/countries.

Discussion

Reduce repetition of results, build a foundation in alignment with existing literature, and draw out conclusions, and implications. E.g. p. 41 the conclusion at the end of paragraph one about a UK study. Draw this out a little more to conclude based on the findings of this study together with existing research. The use of assertive language should help with making more concrete conclusions.

With a little engagement with the HF literature, the authors should present their recommendations for HF implementation. For example, should HF programmes have an emergency/crisis readiness component? Were there failures identified that should be addressed? What kind of encompassing policies would help with HF crisis readiness? How do findings inform policies on harm reduction?

Develop the policy recommendations a bit more by giving examples or comparing proposed changes with existing approaches. For example in Ireland during the pandemic, harm reduction policies were expedited and implemented which meant people in homelessness could access their medication quickly and easily, what policy implications hold for Ontario/Canada?

Wording: throughout the Discussion the authors use the term “technological inequities” and “social inequities”. I would suggest inequalities instead as equity holds meaning as a legal/financial term and inequality or unfairness is more commonly used in the homelessness literature.

Typos

There is a typo in the title, “undeserved”

Pg. 5 line 4, “are is far greater in these…”

Pg. 28 “copying”

Pg. 32 “staffing”

Pg. 35 “clients ‘ “

Error in figure 2. “popultions”

6. PLOS authors have the option to publish the peer review history of their article (what does this mean?). If published, this will include your full peer review and any attached files.

Reviewer #1: No

Reviewer #2: No

---

## [Author Response · Author response to Decision Letter 0]

11 Aug 2022

We thank the reviewers for their helpful feedback, comments and suggestions, which helped to improve the quality, readability and impact of the manuscript. We have provided answer to each reviewer' comment in the enclosed file entitled "Response to Reviewers".

---

## [Decision Letter · Decision Letter 1]

21 Sep 2022

PONE-D-21-25281R1Providing housing first services for an underserved population during the early wave of the COVID-19 pandemic: A qualitative study.PLOS ONE

Dear Dr. Mejia-Lancheros,

Thank you for submitting your manuscript to PLOS ONE. After careful consideration, we feel that it has merit but does not fully meet PLOS ONE’s publication criteria as it currently stands. Therefore, we invite you to submit a revised version of the manuscript that addresses the points raised during the review process. Please submit your revised manuscript by Nov 05 2022 11:59PM. If you will need more time than this to complete your revisions, please reply to this message or contact the journal office at plosone@plos.org. Please include the following items when submitting your revised manuscript:A rebuttal letter that responds to each point raised by the academic editor and reviewer(s). You should upload this letter as a separate file labeled 'Response to Reviewers'.A marked-up copy of your manuscript that highlights changes made to the original version. You should upload this as a separate file labeled 'Revised Manuscript with Track Changes'.An unmarked version of your revised paper without tracked changes. You should upload this as a separate file labeled 'Manuscript'.If applicable, we recommend that you deposit your laboratory protocols in protocols.io to enhance the reproducibility of your results. Protocols.io assigns your protocol its own identifier (DOI) so that it can be cited independently in the future. For instructions see: https://journals.plos.org/plosone/s/submission-guidelines#loc-laboratory-protocols. Additionally, PLOS ONE offers an option for publishing peer-reviewed Lab Protocol articles, which describe protocols hosted on protocols.io. Read more information on sharing protocols at https://plos.org/protocols?utm_medium=editorial-email&utm_source=authorletters&utm_campaign=protocols.

We look forward to receiving your revised manuscript.

Kind regards,

Saeed Shahabi

Academic Editor

PLOS ONE

Journal Requirements:

Reviewers' comments:

Reviewer's Responses to Questions

**Comments to the Author**

1. If the authors have adequately addressed your comments raised in a previous round of review and you feel that this manuscript is now acceptable for publication, you may indicate that here to bypass the “Comments to the Author” section, enter your conflict of interest statement in the “Confidential to Editor” section, and submit your "Accept" recommendation.

Reviewer #1: (No Response)

Reviewer #2: All comments have been addressed

2. Is the manuscript technically sound, and do the data support the conclusions?

Reviewer #1: Yes

Reviewer #2: Yes

3. Has the statistical analysis been performed appropriately and rigorously? 

Reviewer #1: Yes

Reviewer #2: Yes

4. Have the authors made all data underlying the findings in their manuscript fully available?

Reviewer #1: (No Response)

Reviewer #2: No

5. Is the manuscript presented in an intelligible fashion and written in standard English?

Reviewer #1: Yes

Reviewer #2: Yes

6. Review Comments to the Author

Reviewer #1: The authors have sufficiently addressed my feedback and/or provided sufficient justification for their decisions. There remains a number of typos/broken sentences related to the new additions and changes. I would encourage the authors to have a rapid but thorough review to fix these minor concerns.

Reviewer #2: It was a pleasure to revisit this manuscript on HF implementation in Toronto during the COVID-19 pandemic. My feelings are the same that the findings reported here are interesting and important for the field of homelessness research, particularly for HF crisis-readiness. I am satisfied that the authors have worked hard to address my comments (and that of other reviewers). I'd just recommend that the authors proofread the manuscript, as I noticed some typos.

7. PLOS authors have the option to publish the peer review history of their article (what does this mean?). If published, this will include your full peer review and any attached files.

Reviewer #1: No

Reviewer #2: No

---

## [Author Response · Author response to Decision Letter 1]

24 Oct 2022

Response to Editorial and Reviewers’ Comments

We thank the editor and reviewers for their comments and suggestions, which improve the quality and readability of our manuscript. We have highlighted changes in yellow in the marked-up manuscript file. 

Editorial team’s Comments Response

Journal Requirements: 

Please review your reference list to ensure that it is complete and correct. If you have cited papers that have been retracted, please include the rationale for doing so in the manuscript text, or remove these references and replace them with relevant current references. Any changes to the reference list should be mentioned in the rebuttal letter that accompanies your revised manuscript. If you need to cite a retracted article, indicate the article’s retracted status in the References list and also include a citation and full reference for the retraction notic.

Reponse: Thank you for requesting the reference list and ensuring the references are all completed and corrected. We have revised the cited references one by one, correcting any format error or updating any publication details (pages 51-56) 

Reviewers’ Comments

Reviewer #1: 

Comment #1: 

The authors have sufficiently addressed my feedback and/or provided sufficient justification for their decisions. There remain a number of typos/broken sentences related to the new additions and changes. I would encourage the authors to have a rapid but thorough review to fix these minor concerns. 

Response: We thank you for this positive feedback and reassuring comment. As suggested, we have carefully revised and edited the manuscript, addressing existing typos and broken sentences.

Reviewer #2: 

Comment #1: 

It was a pleasure to revisit this manuscript on HF implementation in Toronto during the COVID-19 pandemic. My feelings are the same that the findings reported here are interesting and important for the field of homelessness research, particularly for HF crisis-readiness. I am satisfied that the authors have worked hard to address my comments (and that of other reviewers). I'd just recommend that the authors proofread the manuscript, as I noticed some typos.

Response:Thank you for this comment. We agree with the reviewer on the importance and impact our findings could have on homelessness, housing and pandemic crisis preparedness and response. 

As mentioned above in response to Reviewer #1, we have proofed the manuscript and addressed the remaining typos or editing issues.

---

## [Decision Letter · Decision Letter 2]

17 Nov 2022

Providing housing first services for an underserved population during the early wave of the COVID-19 pandemic: A qualitative study.

PONE-D-21-25281R2

Dear Dr. Mejia-Lancheros,

We’re pleased to inform you that your manuscript has been judged scientifically suitable for publication and will be formally accepted for publication once it meets all outstanding technical requirements.

Kind regards,

Saeed Shahabi

Academic Editor

PLOS ONE

Additional Editor Comments (optional):

Reviewers' comments:

Reviewer's Responses to Questions

**Comments to the Author**

1. If the authors have adequately addressed your comments raised in a previous round of review and you feel that this manuscript is now acceptable for publication, you may indicate that here to bypass the “Comments to the Author” section, enter your conflict of interest statement in the “Confidential to Editor” section, and submit your "Accept" recommendation.

Reviewer #2: All comments have been addressed

2. Is the manuscript technically sound, and do the data support the conclusions?

Reviewer #2: Yes

3. Has the statistical analysis been performed appropriately and rigorously? 

Reviewer #2: Yes

4. Have the authors made all data underlying the findings in their manuscript fully available?

Reviewer #2: No

5. Is the manuscript presented in an intelligible fashion and written in standard English?

Reviewer #2: Yes

6. Review Comments to the Author

Reviewer #2: I have no further comments for the authors. Now that the typos have been addressed it is ready for publication. Well done.

7. PLOS authors have the option to publish the peer review history of their article (what does this mean?). If published, this will include your full peer review and any attached files.

Reviewer #2: No

---

## [Editor Report · Acceptance letter]

23 Nov 2022

PONE-D-21-25281R2 

Providing housing first services for an underserved population during the early wave of the COVID-19 pandemic: A qualitative study. 

Dear Dr. Mejia-Lancheros:

I'm pleased to inform you that your manuscript has been deemed suitable for publication in PLOS ONE. Congratulations! Your manuscript is now with our production department. 

Kind regards, 

on behalf of

Dr. Saeed Shahabi 

Academic Editor

PLOS ONE